# Assessment of Husbandry Practices That Can Reduce the Negative Effects of Exposure to Low Ammonia Concentrations in Broiler Houses

**DOI:** 10.3390/ani12091096

**Published:** 2022-04-23

**Authors:** Leonardo V. S. Barbosa, Daniella J. De Moura, Fernando Estellés, Adrian Ramón-Moragues, Salvador Calvet, Arantxa Villagrá

**Affiliations:** 1College of Agricultural Engineering, State University of Campinas, 501 Candido Rondon Avenue, São Paulo 13083-875, Brazil; 2Institute of Animal Science and Technology, Universitat Politècnica de València, Camino de Vera s.n., 46022 Valencia, Spain; feresbar@upv.es (F.E.); salcalsa@upvnet.upv.es (S.C.); 3Centro de Investigación en Tecnología Animal (CITA), Valencian Institute for Agricultura Research (IVIA), 12400 Segorbe, Spain; armzulu@gmail.com (A.R.-M.); villagra_ara@gva.es (A.V.)

**Keywords:** animal welfare, animal health, immune system, productive parameters, management

## Abstract

**Simple Summary:**

We used two commercial breeds, differing in growth rate: Fast-growing breed and slow-growing breed. We stocked these birds in two different densities. The slow-growing birds was stocked at a high density and the fast-growing birds at a high density and low density. These birds were reared under two different environmental conditions: A control room with a low concentration of ammonia and a second room with a higher concentration. We analyzed management practices such as the effect of ventilation, animal density and growth rate as management possibilities to reduce the negative effect of ammonia on production parameters.

**Abstract:**

Ammonia is an important pollutant emitted by broiler litter that can accumulate inside farms, impairing animal health and welfare productivity. An experiment was designed to evaluate of precision husbandry practices such as the effect of ventilation, animal density and growth rate as management options to reduce the adverse effects of ammonia exposure on productive parameters in broiler houses. Two identical experimental rooms were used in this study. They were programmed to differ in ammonia concentration from day 32 of the growing period (10 and 20 ppm in Room 1 and Room 2, respectively). Three treatments were tested in each room: slow growth in high stocking density (SHD), fast growth in low density (FLD) and fast growth in high density (FHD). Animal weight, feed intake and feed conversion ratio were determined weekly. In addition, the immune status of animals was assessed by weighing the organs related to immune response as stress indicators. Increasing ventilation was effective to control ammonia concentrations. Exposure to ammonia caused no significant effect on productive parameters. However, lowering stocking density improved response to higher ammonia concentrations by lowering the feed conversion ratio. No other relevant effects of differential exposure to ammonia were found in fast-growing animals, either at high or low stocking density. The use of slow-growing breeds had no effect on production parameters. Despite having a slower growth rate, their feed conversion ratio was not different from that of fast-growing breeds. The productive performance of slow-growing animals was not affected by the differential exposure to ammonia, but the reduced spleen size would suggest an impairment of the immune system.

## 1. Introduction

Maintaining a proper rearing environment is essential in broiler production. Ammonia concentration in the air is one of the main factors impairing broiler health, welfare and productivity. Exposure of animals to high ammonia levels causes irritation of mucous membranes and the respiratory tract, conjunctivitis and dermatitis [1,2,3,4]. Animals exposed to ammonia reduce feed consumption and consequently productivity is impaired through lower growth rate, higher mortality and worse feed conversion ratios [5,6,7,8]. It has also been demonstrated that high ammonia concentrations indirectly influence gene expression, affecting the immune response [9] and productivity in terms of slaughter performance and breast yield [10]. For those reasons, welfare regulations establish a concentration threshold for ammonia of 20 ppm according to European Directive 2007/43/CE.

However, the implications of levels below 20 ppm throughout the broiler production cycle are not well known. Most studies do not report relevant effects at concentrations lower than 20 ppm. However, there is evidence that animals’ response starts under lower concentrations. It has been described that concentrations of 15 ppm did not affect growth performance, but induced an anti-inflammatory response in the ileum and altered the tracheal microbiota, causing respiratory tract injury [11,12]. Therefore, understanding the effects of concentrations lower than 20 ppm is necessary. 

Precision husbandry practices are useful tools to reduce the negative impacts of ammonia exposure. More resilient animals, reducing animal density or increasing ventilation rates are effective practices on farms. Growth rate and animal density are related to animal resilience. Slow-growing broilers are healthier and express more behavioral indicators of positive welfare [13]. Additionally, it has been reported that slow growing animals have a greater magnitude of innate immune response to infections [14]. Therefore, broilers with a lower growth rate may be expected to be more robust against ammonia exposure and less affected in terms of welfare, health and productivity. However, to the authors’ knowledge, there is no information on how ammonia exposure affects slow-growing broilers differently from conventional broiler breeds. The effect of stocking density on broiler performance has been described in the literature [15], but the interacting effect of stocking density with the exposure to ammonia is not clear.

Engineering solutions are also available to control ammonia concentrations in broiler farms. The concentration of ammonia in the indoor air is therefore influenced by the diluting effect of ventilation. Ammonia comes from the breakdown of the uric acid excreted by the broilers [16]. The emission rate depends on litter moisture, and litter N content tends to increase as excreta is accumulated in bedding material [17]. Once emitted, the ammonia accumulates inside broiler houses until it is emitted to the atmosphere through the ventilation system. Adjusting ventilation is a strategic mechanism for maintaining an appropriate indoor air quality. It is possible to adjust ventilation to achieve proper indoor air quality, but this normally involves higher energy consumption [18]. Therefore, farmers need to have clear evidence of the welfare and productivity benefits of maintaining ammonia concentrations below certain thresholds.

The hypothesis motivating this study is that there may be additional benefits in reducing stressing factors (ammonia exposure, animal density or growth rate) below the common practice on farm. However, there is scarce information quantifying these effects. Therefore, the objective of this study was to evaluate the influence of two ammonia concentrations (10 and 20 ppm), two broiler breeds (fast vs. medium growth rate) and stocking density (13 vs. 6.5 broiler/m^2^) on the productive performance and physiological parameters during one rearing cycle.

## 2. Materials and Methods

### 2.1. House Description

The experiment was conducted in accordance with the animal research regulations of the EU, with protocol number 2018/VSC/PEA/0067. The test was carried out at the Animal Technology and Research Center (CITA-IVIA), located in Segorbe, (Castellón, Spain). Two identical rooms (Room 1 and Room 2) were used in this trial. Each room had a dimension of 13.2 m × 5.95 m and had independent mechanical ventilation through ceiling fans (Figure 1).

An automated climate control system (DNP Climate Controller, Exafan, Spain) regulated ventilation rates according to commercial breed recommendations of temperature. The environmental temperature was gradually decreased from 32 °C (day 1) to 19 °C (day 42). Temperature was controlled using the climate controller sensor and was recorded together with relative humidity every 10 min using a data logger (HOBO U12, Onsetcomp, Bourne, MA, USA). Additionally, each room was equipped with one electrochemical ammonia sensor (DOL 53, Dräger, Germany). The location of these sensors is presented in Figure 1.

Ammonia was established as a criterion to operate ventilation. In Room 1, ventilation was programmed to maintain a maximum of 10 ppm ammonia, while in Room 2, it was programmed to reach a maximum of 20 ppm. These concentration conditions were programmed to be maintained from the fourth week, that is, in the second half of the production cycle, which is when the ammonia levels are usually higher inside the farms. From the start of the growing period until week 4 of age, both rooms were ventilated following an identical program based on temperature control and animal age. A propane heater was used to maintain adequate room temperatures. Wood shavings (5 cm) were used as the litter material.

The experiment was conducted in winter, when gas concentrations were expected to be higher due to lower ventilation rates. In order to promote NH_3_ emissions and achieve the desired concentrations of ammonia, a urea solution was applied to the litter. Urea was applied manually using a backpack sprayer on days 32, 39, 51 and 56 of the growing period at a dose of 0.21 L m^−2^ of urea solution in distilled water.

### 2.2. Animals and Experimental Design

Two commercial breeds were used, differing in growth rate. The fast-growing breed was Ross®, with a slaughter age of 42 days. The slow-growing breed was Hubbard®, with a slaughter age of 63 days. Three treatments were tested. The first treatment (SHD) was slow-growing birds stocked at a high density (32 kg/m^2^ at slaughter age). The second treatment (FHD) was fast-growing birds at a high density (32 kg/m^2^ at slaughter age). The third treatment (FLD) was fast-growing birds at a final stocking density of 16 kg/m^2^. 

The diet was provided ad libitum. Both commercial breeds received a commercial diet. The feed for the fast-growing breed stocked in high and low density consisted of the Nanta A80, and for the slow growing breed, Nanta A32.

In each room, 18 collective pens were installed to allocate 6 repetitions of each treatment. Each pen was provided with 3 nipple drinkers and a manual feeder and housed 17 birds. The dimensions of high-density pens, corresponding to SHD and FHD treatments, were 1 × 1.3 m^2^, whereas low-density pens (corresponding to FLD) had the same number of animals in double the surface area (2 × 1.3 m^2^). Therefore, in each room, 204 fast-growing and 102 slow-growing birds were housed inside the pens. As the pens only occupied a part of each room, 319 broilers were allocated in the space outside the pens in each room. This was completed to simulate the real ambient conditions inside a broiler house, where ammonia comes from litter. Therefore, a total of 625 birds were used in each room. All birds were fed with commercial starter feed from day 1 to day 18 of testing, and from day 18 until the end of the test, each strain was fed with different feeds according to their demands (Figure 2).

### 2.3. Animal Performance and Health

All the animals housed in the pens were weighed weekly. Feed consumption was also measured in each pen on a weekly basis. The average daily weight gain (DWG) per bird was subsequently calculated for each week of rearing and that accumulated during the test (42 days FLD and FHD and 63 for SHD). The feed conversion rate (FCR) was also calculated for each week and at a cumulative level by dividing the amount of feed consumed by each pen by the growth of all the birds in it.

Thirty animals per treatment were slaughtered on days 21 and 42 to measure the weight of organs related with the immune system. Slow-growing animals were also sampled on day 63. Animals were slaughtered with electric stunning. The immune organs taken from each animal were the spleen, thymus gland, liver and bursa of Fabricius. Organs were weighed using a precision scale (Ohaus Pioneer, PX, Nanikon, Switzerland). Afterwards, weight ratios of each organ were made according to the weight of the birds from which it came to check physiological and anatomical disorders in the immune system organs. On day 63, Fabricio’s bag could not be obtained due to its small size or nonexistence, as this organ tends to disappear as animals grow.

### 2.4. Data Analysis 

Recorded values of ambient temperature, ventilation rate and ammonia concentration were integrated into daily average values presented for descriptive purposes.

The study data showed a normal distribution for the mean test estimates. The statistical analysis of each measured variable was conducted per week of age and accumulated throughout the growing period. A one-way ANOVA analysis was conducted using the software Satgraphics Centurion XVIII, according to the following model:Yijk = μ + Ti + εi
where: Yijk = studied variable (week or accumulated value).

μ = general mean.

Ti = treatment (SHD Room 1, SHD Room 2, FHD Room 1, FHD Room 2, FLD Room 1, FLD Room 2).

εi = residual error.

## 3. Results and Discussion

### 3.1. Control of Environmental Parameters

Environment conditions (temperature, relative humidity, ammonia concentration and ventilation rate) during the experiment are presented in in the Table 1. Since both rooms had an identical number of animals and management, this difference in concentration was promoted through increasing ventilation in Room 1 as compared to Room 2.

Air temperature (Table 1) followed the recommendations for the broiler strains used in this study. Both rooms had a similar temperature during the first four weeks. However, from week 5, the higher ventilation rates in Room 1 caused its average temperature to be reduced by 1.7 °C on average. However, the difference of 1.7 °C was obtained in the general average of all the weeks studied. In the first two weeks, which are the most critical, this difference did not reach 0.3 °C, thus being practically identical in temperature conditions is not enough to cause differences in the performance of the birds. Relative humidity was very similar between rooms. During the four initial weeks, ventilation was kept high, and near-zero ammonia concentration was detected (Table 1). From week 5 until the end of the experiment, the climate control system was operated to create the desired difference in ammonia concentration, averaging 8.27 ppm in Room 1 and 17.1 ppm in Room 2.

### 3.2. Animal Performance

The evolution of animal weight is presented in Table 2. As expected, slow-growing animals had lower weight than the fast-growing ones of the same age from the early stages of the growing period. During the first week, some differences were found among treatments, which could be related to the random choice of animals at the beginning of the experiment. Differences were also found in weeks 3 and 4 between slow-growing animals in both rooms. In those weeks, animals housed in Room 1 had a lower weight than those in Room 2. No more differences were found between slow-growing animals from day 35 until the end of the experiment. For fast-growing broilers, no differences in weight were detected between treatments from day 14 for animals kept at either high or low density.

Table 3 shows feed consumption per treatment and week. Among slow-growing animals, differences between rooms were only found from day 35 to day 42, when animals in Room 2 consumed more feed than in Room 1. No differences were found between rooms for fast-growing animals kept at a high density. However, feed consumption was higher in week 5 for low-density animals in Room 2 compared to Room 1. The accumulated feed consumption at the end of the growing period was higher for slow-growing animals (6.42 and 6.79 in Room 1 and Room 2, respectively) than for fast-growing animals, both at a high density (5.35 and 5.33 in Room 1 and Room 2, respectively) and at a low density (5.03 and 4.71 in Room 1 and Room 2, respectively). In summary, slow-growing animals consumed less feed per week but had higher feed consumption at the end of their growing period since their lifetime was three weeks longer (63 days for slow-growing animals and 42 days for fast-growing animals). Comparing fast-growing animals, those animals kept at a lower density and higher ammonia concentration (Room 2) had lower feed consumption at the end of the growing period.

Table 4 shows the evolution of feed conversion ratio depending on animal age. For the same treatment, no differences were found between Room 1 and Room 2. It was observed that both fast- and slow-growing animals kept at a high density had a similar feed conversion ratio at the end of their growing period. This means a proportional reduction in animal growth rate and feed consumption of slow-growing animals compared to fast-growing animals. However, fast-growing animals kept at a low density had a lower conversion ratio, particularly in those animals subjected to higher ammonia concentrations (Room 2). For the feed conversion ratio accumulated at 28 days of age, values of 1.90 were obtained for SHD, 2.00 for FHD, and 1.96 for FLD stocked in Room 1, and 1.66 for SHD, 2.01 for FHD and 1.82 for FLD stocked in Room 2. At 42 days of age, the following values were obtained: 1.76 for SHD, 2.16 for FHD and 2.09 for FLD stocked in Room 1, and 1.80 for SHD, 2.25 for FHD and 1.96 for FLD stocked in Room 2.

A different exposure of ammonia between Room 1 and Room 2 was achieved using different ventilation management strategies. Although the experiment was planned in winter conditions, where lower ventilation rates are needed, ammonia concentration was negligible until day 32 of the growing period. Litter material remained too dry to generate significant NH_3_ emissions, so on day 32, urea had to be applied in both rooms to increase ammonia production. As the same rate of urea was used in both rooms, it was not expected to disturb the results of the experiment and was effective in producing a controlled amount of ammonia in the rooms. Once ammonia was generated, the differential effect of ventilation originated different ammonia concentrations until the end of the growing period. Therefore, exposure to differential ammonia occurred during the last 10 days for the fast-growing animals and the second half of the growing period for the slow-growing animals. These conditions may be representative of broiler production if new bedding material is used, since concentrations are low during the first weeks of the rearing period [19].

Ventilation operated by the ammonia sensor was effective to control ammonia concentrations. Until day 32, both rooms operated with similar ventilation (less than 2% difference on average). However, from that day, Room 1 had on average 59% higher ventilation than Room 2. This is concurrent with the lower ammonia concentration registered, which was 51% lower in Room 1.

Ventilation has been reported as an effective way to reduce indoor ammonia concentration; this strategy involves more energy consumption. Was r eported that a commercial farm could keep gas concentrations below regulation thresholds (20 ppm) by increasing ventilation, but this would involve increasing electric and heat consumption by 10% and 14%, respectively. Using ventilation to keep lower concentration values would entail higher energy costs as well as instrument maintenance, and these costs would be likely increase exponentially as the target value reduces. Therefore, efforts to keep concentrations lower than those reported in the regulation must be clearly supported in terms of productivity, health and welfare [18].

In this study, we found no significant effect of exposure to ammonia concentration on broilers kept under conventional conditions (fast-growing animals kept at high stocking density). Animals under improved environmental conditions tended to grow faster with similar feed consumption, which resulted in a slightly better conversion rate. These numerical differences are in accordance with the hypothesis of this study. Depending on the stress factor, the organs related to immunity would decrease or increase in size. In the case of stress due to ammonia exposure, the liver, spleen and bursa would be less developed [20]. However, no differences were found in this study.

Exposure to differential ammonia concentrations for only 10 days could be originating that differences are minor and could not be detected in this experiment. The effects of ammonia on animals depend on exposure time [4], and therefore it is necessary to explore the effect of longer exposure times, which could also affect broilers at early stages of their life. 

An ammonia concentration below 20 ppm does not appear to have a clear effect on productive parameters, and similar results have been found in other experiments [11,21]. It seems that exposure to low ammonia concentrations may influence some conditions related to health and welfare [12], but this disturbance is not enough to produce visible changes in performance. According to our results, it seems that the absence of an effect of low ammonia concentrations on productive parameters would also apply for slow-growing strains and animals kept at a lower stocking density.

This study also did not find a significant effect of stocking density on animal growth in fast-growing animals. However, we found that animals reduced feed consumption and improved the feed conversion ratio, particularly for animals kept under higher ammonia concentrations. Improvements of lowering stocking density have been reported in the literature [22,23], but the interacting effect of ammonia concentration has not been reported. It seems that reducing stocking density may facilitate animals responding better to adverse environmental conditions, although the high-density treatment in this study was not sufficiently high to elicit more relevant differences at the animal level, as those obtained in previous studies [15,24].

The difference in productive parameters between slow-growing and fast-growing strains was evident. However, feed conversion rate did not decrease when using the slow-growing strain used in this experiment, which is relevant in terms of efficiency in the use of resources. Slow-growing animals were exposed for more time (31 days) to different ammonia concentrations than fast-growing animals (only 10 days), and therefore more differences would be expected. However, very similar productive parameters were found for slow-growing animals kept either at 10 or 20 ppm. The only difference detected was the relative weight of spleen at the end of the growing period. The lower spleen weight in animals kept at higher ammonia concentrations would support the hypothesis that stressor factors reduce immune organs [20], but this would not be translated into different productive results. 

The results of this study indicate that husbandry practices can improve the conditions of animal growth with respect to the minimum legal requirements. These improved conditions, however, have a relatively low impact on animal performance and therefore their cost effectiveness needs to be further assessed. Results suggest that exposure to low ammonia concentrations (10 vs. 20 ppm) during part of the growing period originates some physiological changes that are not clearly reflected in productive terms, and we did not find evidence that these improved conditions could allow animals to respond better to stressing conditions. This potential effect on animal resiliency needs to be explored by future research.

### 3.3. Weight of the Immune Organs

The effect of stress on the welfare of the animal, its health and therefore its immune system is varied. The spleen, liver, and bursa of Fabricius are used as anatomical indicators of stress [25,26]. The average weight of the lymphoid organs (spleen and bursa of Fabricius) is related to changes in response to stress. The same has been reported for the thymus [27]. In Table 5, we find the evolution of the ratio of the organs that are involved in the immune system and are indicators of stress. Depending on the stress factor, the organ decreases or increases its size. In the case of stress due to a high concentration of NH_3_, the organs are expected to be underdeveloped compared to a normal growth [20]. Contrarily, the expected effect of density is that the immune organs are more developed than normal [15].

Table 5 shows the weight of the organs related to the immune system, expressed in relative terms with respect to total animal weight. For fast-growing animals, information is available until the slaughter day (42). For slow-growing animals, the bursa of Fabricius was too small, and it could not be measured at day 63, so no information is provided.

Spleen weight was very variable on day 21 of the growing period and no relevant differences were found on day 42 in absolute weight. However, as shown in Table 5, the spleen had a higher weight when expressed in relative terms to animal weight due to the lower weight of slow-growing animals at the same age. Additionally, animals at a higher ammonia concentration tended to have a lower spleen weight than those at lower concentrations. At slaughter age, slow-growing animals’ spleens were heavier in animals kept at lower ammonia concentration (Room 1) compared to those at higher ammonia concentrations (Room 2). The relative weight of the thymus gland was also affected by animal strain and slow-growing animals had a higher relative weight than fast-growing animals. No differences for stocking density were detected due to ammonia concentration. A similar situation was found for the liver and bursa of Fabricius, where no clear effect was found because of changing exposure to ammonia concentration or stocking density.

Studies show that in situations of stress due to exposure to ammonia, the liver, spleen and bursa would be less developed [20,28,29]. This study found some changes in the size of immune organs, so it is suggested that more in-depth studies are necessary in this direction to confirm if in fact the larger size of the organs are correlated with situations of greater stress such as higher ammonia concentration and higher stocking density.

## 4. Conclusions

This study evaluated the effect of husbandry practices on broiler response to low ammonia concentrations. Ventilation was used to generate exposure to different ammonia concentration in the final part of a growing period (approximately 10 and 20 ppm). 

Reducing ammonia concentrations had a low influence on productive parameters regardless of the animal strain (slow- or fast-growing animals) or stocking density (32 vs. 16 kg m^−2^ at slaughter age) was tested. However, animals kept at a lower stocking density had a better conversion rate than those under high density, particularly at higher ammonia exposure. 

Slow-growing animals were exposed to an ammonia concentration for a longer time than fast-growing animals. No effects on growth parameters were found, but the increased weight of immune organs (spleen) suggests a deleterious effect on the immune response. However, despite being an indication, it is not possible to say that there is an immunological dysfunction based only on the weight of the lymphoid organs. Therefore, we recommend that further research be carried out in this direction.

Husbandry practices improving animal conditions beyond the minimum legal requirements seem to have a low effect on productivity.

## Figures and Tables

**Figure 1 animals-12-01096-f001:**
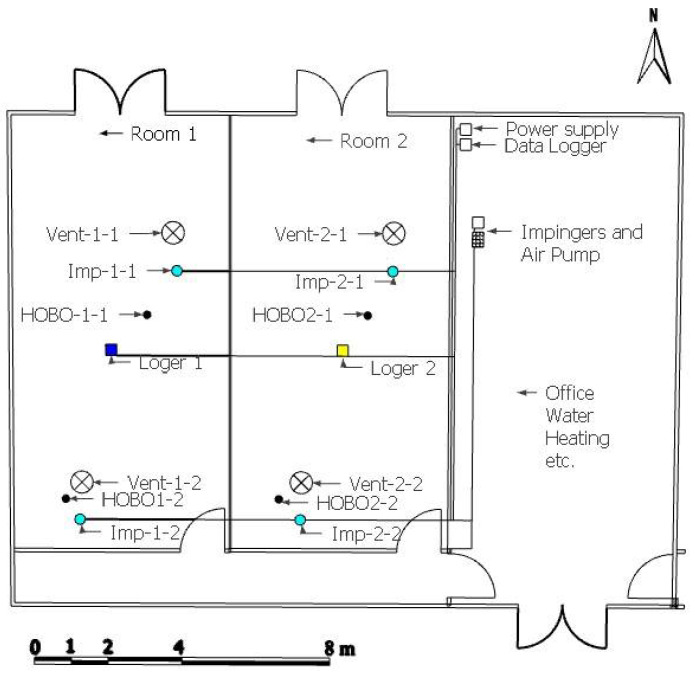
Top view of broiler house. Location of fans (Vent), temperature and relative humidity loggers (HOBO), and ammonia sensors is indicated.

**Figure 2 animals-12-01096-f002:**
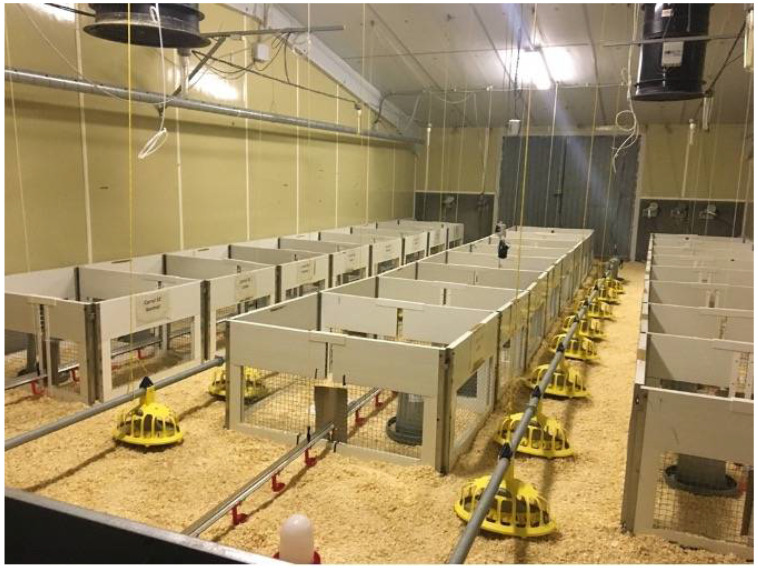
Experimentation room with details of the layout of the experimental compartments.

**Table 1 animals-12-01096-t001:** Weekly average values of temperature (°C), relative humidity (%) ventilation rate (m^3^ h^−1^ animal^−1^) and ammonia concentration (ppm) in the broiler houses (Room 1 and Room 2).

	Temperature (°C)		Relative Humidity (%)		Ventilation Rate (m^3^ h^−1^ animal^−1^)		NH_3_ Concentration (ppm)	
	Room 1	Room 2	Room 1	Room 1	Room 1	Room 2	Room 1	Room 2
Week 1	33.1	33.0	22.1	22.9	0.8	0.3	0.0	0.0
Week 2	29.8	29.5	29.4	31.7	4.2	3.7	0.0	0.0
Week 3	27.0	26.3	29.1	31.5	5.5	5.8	0.0	0.0
Week 4	24.7	23.8	29.5	30.5	6.5	7.7	0.0	0.0
Week 5	21.9	22.8	46.4	50.7	3.9	3.5	0.3	6.5
Week 6	20.3	22.0	54.1	59.2	3.5	2.5	4.5	19.0
Week 7	19.7	22.2	55.9	61.1	5.0	2.6	9.0	20.4
Week 8	20.4	21.7	48.4	50.8	6.5	3.7	12.5	14.4
Week 9	19.9	21.5	51.1	54.7	6.8	4.4	10.1	16.1

**Table 2 animals-12-01096-t002:** Evolution of animal weight (g per bird) during the rearing period (*n* = 30 animals per treatment).

**Age**	**Treatment SHD**		**Treatment FHD**		**Treatment FLD**		**S.E.**	** *p* ** **-Value**
**(days)**	Room 1	Room 2	Room 1	Room 2	Room 1	Room 2		
7	134.4 ^a^	144.8 ^ab^	164.4 ^d^	151.1 ^bc^	160.4 ^cd^	160.5 ^cd^	3.9	≤0.05
14	338.8 ^a^	307.4 ^a^	390.3 ^b^	436.1 ^b^	419.5 ^b^	437.7 ^b^	16.6	≤0.05
21	525.8 ^a^	563.3 ^b^	775.4 ^c^	801.1 ^c^	780.6 ^c^	797.7 ^c^	11.9	≤0.05
28	853.8 ^a^	960.8 ^b^	1242.7 ^c^	1226.5 ^c^	1239.2 ^c^	1227.9 ^c^	29.8	≤0.05
35	1294.1 ^a^	1302.2 ^a^	1840.8 ^b^	1779.3 ^b^	1825.9 ^b^	1750.7 ^b^	35.2	≤0.05
42	1769.0 ^a^	1801.1 ^a^	2527.3 ^b^	2415.0 ^b^	2475.6 ^b^	2446.6 ^b^	53.8	≤0.05
49	2192.0 ^a^	2276.3 ^a^	-	-	-	-	69.1	0.40
56	2676.8 ^a^	2776.0 ^a^	-	-	-	-	94.8	0.48
63	3127.7 ^a^	3275.2 ^a^	-	-	-	-	103.0	0.35

Different letters within a row indicate statistically significant differences among treatments. SHD = slow growing and high density; FHD = fast growing and high density; FLD = fast growing and low density. S.E. = standard error. Room 1 was programmed to reach maximum 10 ppm NH_3_ and Room 2 to maximum 20 ppm NH_3._

**Table 3 animals-12-01096-t003:** Evolution of feed consumption (g per animal and week) during the rearing period (*n* = 30 animals per treatment).

**Age**	**Treatment SHD**		**Treatment FHD**		**Treatment FLD**		**S.E.**	** *p* ** **-Value**
**(days)**	Room 1	Room 2	Room 1	Room 2	Room 1	Room 2		
7	0.116 ^a^	0.113 ^a^	0.136 ^b^	0.132 ^b^	0.132 ^b^	0.128 ^b^	0.004	≤0.05
14	0.256 ^a^	0.255 ^a^	0.368 ^b^	0.373 ^b^	0.350 ^b^	0.370 ^b^	0.008	≤0.05
21	0.518 ^a^	0.490 ^a^	0.732 ^b^	0.693 ^b^	0.703 ^b^	0.627 ^b^	0.037	≤0.05
28	0.672 ^a^	0.664 ^a^	1.155 ^b^	1.184 ^b^	1.135 ^b^	1.038 ^b^	0.068	≤0.05
35	0.677 ^a^	0.682 ^a^	1.452 ^c^	1.445 ^c^	1.430 ^c^	1.114 ^b^	0.048	≤0.05
42	0.814 ^a^	0.978 ^b^	1.506 ^d^	1.506 ^d^	1.287 ^c^	1.438 ^cd^	0.056	≤0.05
49	0.976 ^a^	1.032 ^a^	-	-	-	-	0.030	0.22
56	1.155 ^a^	1.242 ^a^	-	-	-	-	0.050	0.20
63	1.236 ^a^	1.333 ^a^	-	-	-	-	0.050	0.19

Different letters within a row indicate statistically significant differences among treatments. SHD = slow growing and high density; FHD = fast growing and high density; FLD = fast growing and low density. S.E. = standard error. Room 1 was programmed to reach maximum 10 ppm NH_3_ and Room 2 to maximum 20 ppm NH_3_.

**Table 4 animals-12-01096-t004:** Evolution of feed conversion ratio per week during the rearing period (*n* = 30 animals per treatment).

**Age**	**Treatment SHD**		**Treatment FHD**		**Treatment FLD**		**S.E.**	** *p* ** **-Value**
**(days)**	Room 1	Room 2	Room 1	Room 2	Room 1	Room 2		
7	1.21 ^ab^	1.07 ^a^	1.12 ^ab^	1.27 ^b^	1.13 ^ab^	1.10 ^a^	0.05	0.10
14	1.56 ^b^	1.59 ^b^	1.38 ^a^	1.32 ^a^	1.36 ^a^	1.39 ^a^	0.05	≤0.05
21	2.22 ^b^	1.94 ^ab^	2.11 ^ab^	1.92 ^ab^	2.01 ^ab^	1.77 ^a^	0.12	0.17
28	2.01 ^ab^	1.74 ^a^	2.53 ^c^	2.79 ^c^	2.53 ^bc^	2.39 ^bc^	0.18	≤0.05
35	1.60 ^a^	2.06 ^ab^	2.44b ^c^	2.63 ^c^	2.45 ^bc^	2.09 ^ab^	0.18	≤0.05
42	1.72 ^a^	2.06 ^ab^	2.20 ^b^	2.30 ^b^	2.02 ^ab^	2.16 ^b^	0.12	≤0.05
49	2.46 ^a^	2.28 ^a^	-	-			0.26	0.63
56	2.39 ^a^	2.51 ^a^	-				0.07	0.25
63	2.76 ^a^	2.69 ^a^	-				0.11	0.66

Different letters within a row indicate statistically significant differences among treatments. SHD = slow growing and high density; FHD = fast growing and high density; FLD = fast growing and low density. Room 1 was programmed to reach maximum 10 ppm NH^3^ and Room 2 to maximum 20 ppm NH_3_.

**Table 5 animals-12-01096-t005:** Evolution of the relative weight of spleen, thymus, liver and bursa Fabricius (g per 1000 g animal weight) during the rearing period.

**Organ**	**Age**	**Treatment SHD**		**Treatment FHD**		**Treatment FLD**		***p*-Value**
	**(days)**	Room 1	Room 2	Room 1	Room 2	Room 1	Room 2	
Spleen	21	1.11 ± 1.17	1.27 ± 1.24	3.82 ± 1.19	0.94 ± 1.17	0.91 ± 1.19	0.94 ± 1.17	0.47
	42	1.62 ^b^ ± 0.06	1.46 ^b^ ± 0.06	1.25 ^a^ ± 0.06	1.18 ^a^ ± 0.06	1.46 ^b^ ± 0.06	1.21 ^a^ ± 0.06	≤0.05
	63	1.26 ^b^ ± 0.05	1.10 ^a^ ± 0.05	-	-	-	-	≤0.05
Thymus	21	2.88 ^c^ ± 0.15	2.98 ^c^ ± 0.16	1.85 ^ab^ ± 0.16	2.04 ^b^ ± 0.15	1.75 ^ab^ ± 0.16	1.57 ^a^ ± 0.15	≤0.05
	42	2.90 ^b^ ± 0.16	2.71 ^b^ ± 0.16	1.59 ^a^ ± 0.16	1.80 ^a^ ± 0.16	1.81 ^a^ ± 0.16	1.79 ^a^ ± 0.16	≤0.05
	63	2.44 ± 0.16	2.22 ± 0.16	-	-	-	-	0.32
Liver	21	29.89 ^c^ ± 0.79	29.38 ^bc^ ± 0.83	26.69 ^a^ ± 0.80	27.71 ^abc^ ± 0.79	27.50 ^ab^ ± 0.80	28.35 ^abc^ ± 0.79	≤0.05
	42	20.55 ^a^ ± 0.48	21.69 ^a^ ± 0.48	20.83 ^a^ ± 0.48	21.56 ^a^ ± 0.48	21.76 ^a^ ± 0.48	21.25 ^a^ ± 0.48	0.38
	63	17.68 ± 0.52	18.69 ± 0.52	-	-	-	-	0.18
Bursa of Fabricius	21	2.85 ^b^ ± 0.13	2.33 ^a^ ± 0.14	1.98 ^a^ ± 0.13	2.09 ^a^ ± 0.13	2.12 ^a^ ± 0.13	2.33 ^a^ ± 0.13	≤0.05
	42	0.54 ^bc^ ± 0.05	0.44 ^abc^ ± 0.05	0.42 ^ab^ ± 0.05	0.45 ^abc^ ± 0.05	0.40 ^a^ ± 0.05	0.56 ^c^ ± 0.05	0.08
	63	-	-	-	-	-	-	-

Different letters within a row indicate statistically significant differences among treatments. SHD = slow growing and high density; FHD = fast growing and high density; FLD = fast growing and low density. Room 1 was programmed to reach maximum 10 ppm NH_3_ and Room 2 to maximum 20 ppm NH_3_.

## Data Availability

Not applicable.

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
