# Peer review of "Assessment of Husbandry Practices That Can Reduce the Negative Effects of Exposure to Low Ammonia Concentrations in Broiler Houses"

_animals, 2022, doi:10.3390/ani12091096_

Round 1

Reviewer 1 Report

The paper, "Assessing Precision Feeding Practices to Reduce the Negative Effects of Ammonia Concentrations in Broiler Houses," aims to explore the effects of animal density and ammonia concentration on broiler production parameters. It was found that reducing stocking density improved the response to higher ammonia concentrations by reducing feed conversion. The writing is smooth and well-designed. But there are still some areas for improvement. It is recommended to publish with major revisions.

  1. What is the basis for the selection of ammonia concentration? Ammonia exposure time?
  2. Fast and slow growth uses different breeds of chickens, which in my opinion does not control for variables. Please explain the reasons for choosing these two varieties.
  3. Please provide the feed ratio, source, and manufacturer for feeding.
  4. Why wasn't slow-growing low-density rearing established?
  5. Have all the data in this study been tested for normal distribution?
  6. The description of the experimental results does not show the size of the significant P value.
  7. The data involved in the Tables do not state units.
  8. This experiment has not done any detection of the mechanism and immune level. I think it is unreasonable to say that the immune dysfunction is abnormal only based on the weight of the immune organs.
  9. In 3.3 The weight of immune organs, there is a long discussion. I think this part of the discussion does not belong to this part. It is suggested that the results and discussions be divided into two parts. And add some references to the literature to enrich the discussion section. Or you can choose to spread the long discussion after each corresponding result.
  10. Please note the unit abbreviation in the text. For example: "minutes" is changed to "min"
  11. Please pay attention to the writing format of numbers and units in the whole text, such as: "13.2m" to "13.2 m", "32º C" to "32 º C".

Author Response

Response to the Academic Editor's comments

manuscript (animals-1635108)

The authors thank you and the reviwers for the comments that made the manuscript much more appropriate and focused.

1- What is the basis for the selection of ammonia concentration? Ammonia exposure time?

Answer: The basis for choosing the ammonia concentration was establishing a maximum of 20ppm as recommended by the European Union as safe levels for broilers. However, we would like to know the effects on performance at lower concentrations such as up to 10ppm and compare with higher concentrations (maximum of 20ppm concentration). The time of exposure to ammonia was variable, since the solubilized urea was sprinkled on the litter, there was a peak of ammonia concentration in the environment and then it dropped. In room 1 (control room) ammonia was not sprayed. The ammonia concentration was not maintained constantly, it was only controlled so that it did not exceed the stipulated maximum levels and remained. If the concentration exceeded 10 ppm in room 1 and 20 ppm in room two, the exhaust fans were activated and the air renewal rate increased.

2- Fast and slow growth uses different breeds of chickens, which in my opinion does not control for variables. Please explain the reasons for choosing these two varieties.

Answer: The reason to choose these two strains would be to verify if, because the slow-growing strains are less physiologically challenged by an accelerated growth, they would present a better performance against those of fast growth in an environment with ammonia as a stressor. That is, they would suffer less from the effects of ammonia compared to those with rapid growth.

3- Please provide the feed ratio, source, and manufacturer for feeding.

Answer: The diet was provided ad libitum. Both strains received a commercial diet. The feed for the fast growing strain housed in high and low density consisted of the Nanta A80 weight gain ration and for the slow growing strain Nanta A32.

Added on Line 133-135

4- Why wasn't slow-growing low-density rearing established?

Answer: The simulation aimed to follow conditions more similar to those found at farm levels. If we put a low density of housing for slow-growing birds, we would have an animal mass density of birds that is not representative and little ammonia emission, unlike what happens in real production conditions. Densities were based on kilogram live weight/m2, if we had a slow growing treatment at low density we would have few kilogram live weight/m2.

5- Have all the data in this study been tested for normal distribution?

Answer: The study data showed a normal distribution for the mean test evaluations.

Line: 169

6- The description of the experimental results does not show the size of the significant P value.

Answer: The P-value was added.

7- The data involved in the Tables do not state units.

Answer: The units were indicated.

8- This experiment has not done any detection of the mechanism and immune level. I think it is unreasonable to say that the immune dysfunction is abnormal only based on the weight of the immune organs.

Answer: We agree that we cannot make this statement but we believe that the change in the weight of immune organs can be better investigated in more depth because it may signal an alteration in the immune system of birds.

9- In 3.3 The weight of immune organs, there is a long discussion. I think this part of the discussion does not belong to this part. It is suggested that the results and discussions be divided into two parts. And add some references to the literature to enrich the discussion section. Or you can choose to spread the long discussion after each corresponding result.

Answer: We agreed to spread the discussion.

10- Please note the unit abbreviation in the text. For example: "minutes" is changed to "min"

Answer: Changed to “min”

11- Please pay attention to the writing format of numbers and units in the whole text, such as: "13.2m" to "13.2 m", "32º C" to "32 º C".

Answer: They have been corrected as recommended.

Reviewer 2 Report

First of all, it is important to mention that the subject of this manuscript fits perfectly within the scope of the Special Issue. The authors intended to verify whether low ammonia concentrations, below the ammonia threshold of 20 ppm established by the European Directive 2007/43/EC, continued to exert any effect on broilers subjected to different management conditions. This goal is very laudable because it could generate results with a strong impact on the poultry industry, leading to a rethink of the regulatory thresholds imposed by the European authorities.

Despite the above, I found some more questionable aspects that I point out in the following lines.

First, the title should focus more on the possibility that even low levels of ammonia can affect broilers health.

Using of a fourth group in the test, i.e. slow growth in low stocking density, might have helped to confirm the hypothesis that ammonia may have a greater impact on high density groups. Also, I do not find a strong justification for not including this group in the study, which could function as a "control", even due to the longer exposure time to ammonia when comparing to those broilers with fast growth.

In the Discussion, the potential impact of lower Room 1 temperature (reduced in 1.7 °C on average) on production parameters is not addressed.

The weighing of immunity-related organs to assess the impact of ammonia on animals, the core of this study, is based on a single reference (#20 Wang et al 2010) and there is not much else to support this methodology in the manuscript.

Author Response

Response to the Academic Editor's comments

manuscript (animals-1635108)

The authors thank you and the reviwers for the comments that made the manuscript much more appropriate and focused.

First of all, it is important to mention that the subject of this manuscript fits perfectly within the scope of the Special Issue. The authors intended to verify whether low ammonia concentrations, below the ammonia threshold of 20 ppm established by the European Directive 2007/43/EC, continued to exert any effect on broilers subjected to different management conditions. This goal is very laudable because it could generate results with a strong impact on the poultry industry, leading to a rethink of the regulatory thresholds imposed by the European authorities.

Despite the above, I found some more questionable aspects that I point out in the following lines.

First, the title should focus more on the possibility that even low levels of ammonia can affect broilers health.

Answer: The title was updated according to this comments as follows:

“Evaluation of precision husbandry practices to reduce the negative effects of EXPOSURE TO LOW ammonia concentrations in broiler houses”

Using of a fourth group in the test, i.e. slow growth in low stocking density, might have helped to confirm the hypothesis that ammonia may have a greater impact on high density groups. Also, I do not find a strong justification for not including this group in the study, which could function as a "control", even due to the longer exposure time to ammonia when comparing to those broilers with fast growth.

Answer: The simulation aimed to follow conditions more similar to those found at field levels. If we put a low density of housing for slow-growing birds, we would have a density of birds that is not representative and little ammonia emission, unlike what happens in real production conditions. Densities were based on kilogram live weight/m2, if we had a slow growing treatment at low density we would have few kilogram live weight/m2.

In the Discussion, the potential impact of lower Room 1 temperature (reduced in 1.7 °C on average) on production parameters is not addressed.

Answer: The difference of 1.7 °C was on average. In the first two weeks, which are most critical, this difference did not reach 0.3 °C, that is, practically identical in temperature conditions, not being enough to cause differences in the performance of the birds.

This is explained in lines: 188-191

The weighing of immunity-related organs to assess the impact of ammonia on animals, the core of this study, is based on a single reference (#20 Wang et al 2010) and there is not much else to support this methodology in the manuscript.

Answer: New references were inserted to support the methodology

Round 2

Reviewer 1 Report

The modification of the article is reasonable, the results data support the conclusion, and the modification is reasonable, so it is recommended to accept the article.

Author Response

Response to the Academic Editor's comments

Manuscript (animals-1635108)

The authors thank you and the reviwers for the comments that made the manuscript much more appropriate and focused. Thank you very much for ordering the article to be published.

Reviewer 2 Report

I still feel that the title does not fit the essence of this manuscript. I suggest something like: “Assessment of husbandry practices that can reduce the negative effects of exposure to low ammonia concentrations in broiler houses”.

Despite understanding the authors' explanations regarding the non-inclusion of a fourth group in the test, i.e. slow growth in low stocking density, as it does not represent real production conditions, I still believe that this might have helped to confirm the hypothesis that even small concentrations of ammonia may have a great impact on high density groups. Testing hypotheses in scientific studies implies, above all, controlling variables, regardless of whether or not they replicate real production conditions.

Assuming that there is an immune dysfunction based only on the weight of the lymphoid organs is quite debatable, because it can vary for reasons other than exposure to ammonia.

In the first paragraph of 3.3 (Weight of the immune organs, lines 325-334), there is some rationale for using lymphoid organs based on references #1 to #5. However, I read all three references to broilers (#1, #4 and #5) and I didn't find a single reference to the impact of ammonia on lymphoid organs! The other two references (#2 and #3) are about human exposure to ammonia (poultry workers and hairdressers). Also, although the authors mention that "New references were inserted to support the methodology", I did not find these new references in the revised manuscript. As it is one of the main conclusions of this study, it should be better supported by the existing literature.

Author Response

Response to the Academic Editor's comments

Manuscript (animals-1635108)

The authors thank you and the reviwers for the comments that made the manuscript much more appropriate and focused.

I still feel that the title does not fit the essence of this manuscript. I suggest something like: “Assessment of husbandry practices that can reduce the negative effects of exposure to low ammonia concentrations in broiler houses”.

Answer: Changed as per reviewer recommendation 

Despite understanding the authors' explanations regarding the non-inclusion of a fourth group in the test, i.e. slow growth in low stocking density, as it does not represent real production conditions, I still believe that this might have helped to confirm the hypothesis that even small concentrations of ammonia may have a great impact on high density groups. Testing hypotheses in scientific studies implies, above all, controlling variables, regardless of whether or not they replicate real production conditions.

Answer: We agree with your position, however we are not able to obtain this data for this current study because it has already been fully carried out. We consider that the data from this study contribute with interesting information, but we do not rule out the possibility of future studies considering the low stocking density fort slow-growth birds.

Assuming that there is an immune dysfunction based only on the weight of the lymphoid organs is quite debatable, because it can vary for reasons other than exposure to ammonia.

Answer: We agree, we made changes to the text and conclusion. Despite being an indication of a diffusion in the immune system, we cannot say based only on weight.

Line: 369-3673

Line: 395-397 

In the first paragraph of 3.3 (Weight of the immune organs, lines 325-334), there is some rationale for using lymphoid organs based on references #1 to #5. However, I read all three references to broilers (#1, #4 and #5) and I didn't find a single reference to the impact of ammonia on lymphoid organs! The other two references (#2 and #3) are about human exposure to ammonia (poultry workers and hairdressers). Also, although the authors mention that "New references were inserted to support the methodology", I did not find these new references in the revised manuscript. As it is one of the main conclusions of this study, it should be better supported by the existing literature.

Answer: We authors are very sorry. By some mistake the references of line 342-351 do not update automatically in the previous revision. They are now updated.
